# Structural basis for carbohydrate recognition by the Gal/GalNAc lectin of *Entamoeba histolytica* involved in host cell adhesion

Samuel F. Gérard[1,2], Christina Redfield[1], Matthew K. Higgins [1,2]*

1 Department of Biochemistry, University of Oxford, Oxford, United Kingdom, 2 Kavli Institute for Nanoscience Discovery, University of Oxford, Oxford, United Kingdom

* matthew.higgins@bioch.ox.ac.uk

## Abstract

Intestinal amoebiasis is caused by *Entamoeba histolytica,* one of the deadliest human-infective parasites. Central to its pathogenicity is its binding to mucosal carbohydrates, which precedes tissue damage by trogocytosis. Carbohydrate binding is mediated by a single adhesin, the galactose/N-acetylgalactosamine (Gal/GalNAc) lectin, which is the leading vaccine candidate for amoebiasis. We present the structure of the native heterodimeric lectin, revealing an ordered core containing the light chain and the N-terminal region of the heavy chain. Structures obtained in the presence of ligand show that the Gal/GalNAc binding site is in the light chain, which adopts a β-trefoil fold found in other lectins. An elongated arm emerges from the heavy chain, which adopts multiple positions and may be modulated by sugar binding. This study reveals the molecular basis for sugar binding by the *Entamoeba histolytica* Gal/GalNAc lectin, a prerequisite for parasite invasion and development of intestinal disease.

## Author summary

*Entamoeba histolytica* is the second most deadly eukaryotic parasite to affect humans. It causes amoebiasis, a disease in which the parasite induces dysentery and diarrhoea. If untreated, the infection can then develop into tissue damage, with the amoeba inducing abscesses in tissues such as the liver. These are caused when it 'bites' bits off our cells, through a process known as trogocytosis. An important molecular complex lies at the centre of this pathology. A lectin is found on the surface of the amoeba, which binds to carbohydrate-containing molecules on the surfaces of our cells. These interactions are required for the amoeba to attack our tissues and vaccination with parts of the lectin can prevent tissue damage. In this study, we show for the first time what this lectin looks like and how it recognises carbohydrates. We also show that the lectin contains an extended lever arm, which can adopt multiple positions relative to the ordered

**Data availability statement:** Crystallography data is deposited at the Protein Data Bank with access codes 9GED, 9GEE, 9GEG and 9GEH for HgL_03 alone and in the presence of Gal, GalNAc or LacNAc respectively. Electron microscopy data is deposited at the Protein Data Bank with accession codes 9GJA-9GJI and in the Electron Microscopy Data Base with accession codes EMDB-51385 (lectin mode 1), EMDB-51386 (lectin mode 2), EMDB-51387 (lectin mode 3), EMDB-51392 (lectin with Gal mode 1), EMDB-51393 (lectin with Gal mode 2), EMDB-51394 (lectin with Gal mode 3), EMDB-51395 (lectin with LacNAc mode 1), EMDB-51396 (lectin with LacNAc mode 2) and EMDB-51397 (lectin with LacNAc mode 3). All other data is included with the manuscript as source data.

**Funding:** This work was funded by the Wellcome Trust. MKH is a Wellcome Investigator (220797/Z/20/Z) and SFG was funded by the graduate program in Cellular Structural Biology (218482/Z/19/Z). The funders had no role in study design, data collection and analysis, decision to publish, or preparation of the manuscript.

**Competing interests:** The authors have declared that no competing interests exist.

core. Future studies will reveal what the lever arm does and how antibodies against it prevent killing of our cells. In the meantime, this first view of the lectin shows us how this deadly amoeba recognises our cells and will guide vaccine development.

## Introduction

*Entamoeba histolytica* is one of the most prevalent and deadly intestinal pathogens to affect humans [1,2]. This unicellular amoeba causes amoebiasis, a disease associated with diarrhoea, amoebic dysentery, and colitis, while extra-intestinal infections can cause amoebic liver abscesses [3]. Infection often begins when amoebas are ingested with food or water contaminated with mature cysts [3]. These develop into trophozoites which colonise the large intestine, where they feed on bacteria and food particles. While most remain largely commensal, in around 10% of infections they become pathogenic, degrading the mucosal layer, and killing exposed epithelial cells, causing formation of flask-shaped ulcers [4]. In some cases, they also enter the blood, reaching secondary sites of infection such as the liver, where ulcers also form.

During early steps of invasion, trophozoites adhere to mucosal carbohydrates via a surface galactose/N-acetylgalactosamine (Gal/GalNAc) lectin [5]. This heterodimer consists of a 170-kDa 'heavy' chain (HgL), with a single transmembrane helix close to the C-terminus [6], linked to a 31/35-kDa 'light' chain (LgL) through a disulphide bond-containing interface [6], each expressed from a different gene. Gal/GalNAc-containing sugars or soluble lectin protein prevent adherence [7–10] and cells which lack O-linked Gal/GalNAc moities are also resistant to killing and cytolysis [11,12], suggesting that the carbohydrate binding function of the lectin is essential for amoebas to bind to host cells and to mediate contact-dependent killing.

The lectin is specific for terminal Gal and GalNAc moieties of mucin O-glycans [13]. Binding to Gal and GalNAc has been attributed to a 104-residue 'carbohydrate recognition domain' (CRD) contained in the cysteine-rich region of HgL [14]. However, the molecular and structural details of how carbohydrate binding occurs remain elusive. In addition, there is currently no structure available for the heterodimeric Gal/GalNAc lectin, and its large size and different reported roles suggest that it most likely has other functions in addition to carbohydrate binding.

The lectin is also a promising vaccine candidate. Acquired resistance to infection in children is associated with mucosal IgA against the lectin [15]. Indeed, antibodies which target the lectin prevent a wide range of pathological processes, including adherence to mucins, contact-dependent cytotoxicity, and resistance to complement [16], while gerbils immunised with native lectin are protected from developing amoebic liver abscesses [17]. Indeed, multiple trials of lectin-based vaccines have already been performed in animals [18–20].

In this study, we present the crystal structure of the reported CRD and cryo-electron microscopy (cryo-EM)-derived structures of the native Gal/GalNAc lectin. Using saturation transfer difference nuclear magnetic resonance (STD NMR) we

show that binding to Gal/GalNAc based glycans is not mediated by the previously reported CRD or the full ectodomain of HgL. Instead, we find that LgL is a ricin-type lectin which contains the Gal/GalNAc binding site. LgL associates with the ordered N-terminal core of HgL, while the C-terminal part of HgL forms an elongated arm that adopts multiple conformations. These findings reveal the molecular mechanism of sugar binding by the lectin and the central role of the LgL subunit, whose function has been largely overlooked.

## Results

### Structural studies and STD NMR show that the previously described carbohydrate recognition domain does not bind Gal/GalNAc

Previous studies have indicated that a lectin fragment, known as the carbohydrate recognition domain (CRD, residues 895–998) contains the Gal and GalNAc binding site [14]. We therefore aimed to express the CRD, as well as other larger fragments of the HgL subunit. As HgL is cysteine-rich and glycosylated, we used a mammalian expression system. A protein matching the domain boundaries of the CRD failed to express. Instead, we produced a set of proteins, guided by AlphaFold2 [21,22] modelling of domain boundaries, which contained the CRD (Fig 1A). Where AlphaFold2 predicted free disulphide bonds in these constructs, due to exposure of cysteines normally forming disulphide bonds with parts of the protein removed, these were mutated to serines to prevent aggregation. The smallest CRD-containing protein to express in a soluble, folded form was HgL_03 (807–992, Cys811Ser Cys972Ser).

To investigate Gal/GalNAc binding, we obtained crystals of HgL_03 that diffracted to 2.48 Å. The structure, solved by molecular replacement, reveals a novel fold (Fig 1B and S1 Table), as no analogous structure was found in the Protein Data Bank using DALI search. Notably there was no structural similarity to any previously characterised lectins. The N-terminal domain (residues 825–918) is globular and contains two pairs of antiparallel β-strands flanked by three small helices. This is followed by three small modules (919–992) which form an elongated arrangement. A pair of antiparallel β-strands (923–939) is followed by a central module consisting of a helix and a pair of antiparallel β-strands (940–962), which is linked by a 13-residue linker to a further pair of antiparallel β-strands (976–991). A total of fourteen cysteines form seven disulphide bonds, two of which join adjacent modules to one another, thereby fixing their relative orientations (Fig 1B). No such disulphide bond is observed in the final module, potentially allowing hinge movement at this location.

As HgL_03 contains all but six residues from the previously described CRD, we next aimed to understand Gal/GalNAc binding by co-crystallising it in the presence of either 250 mM Gal, 250 mM GalNAc or 50 mM N-acetyllactosamine (LacNAc), and obtained electron density maps at 1.82 Å, 1.85 Å and 2.32 Å, respectively. These were solved by molecular replacement and difference maps were produced by subtracting the electron density for the apo structure from that obtained in the presence of each ligand (Fig 1C). In none of the three cases were we able to detect the presence of any ligand.

In the absence of binding of HgL_03 to Gal/GalNAc-containing glycans in our crystallisation conditions, we next investigated carbohydrate binding by STD NMR in a physiological buffer, as this is a method suited to the study of ligands with medium to weak affinity since it relies on fast exchange between bound and free states of the ligand. This was expected to be suitable for analysis of the CRD, which is reported to bind to Galβ1–4 GalNAc with Kd~2.3 µM [23]. In addition to studying HgL_03, we also studied recombinantly expressed full HgL ectodomain and three other fragments (Fig 1A) encompassing the CRD. We were unable to recombinantly express three different isoforms of LgL (EHI_035690, EHI_148790, EHI_183400) in either bacterial, insect cell or mammalian cell systems and were therefore unable to measure the binding of the light chain to carbohydrates. Proteins were saturated in the aromatic region, where no signal for carbohydrate was observed, and STD signals were measured for each protein in the presence of Gal. While a strong STD signal was obtained for positive control protein Jacalin, with each peak assigned to the protons of Gal, no STD signal could be detected in the same conditions in the presence of HgL_03 or any of the other HgL fragments (Fig 1D). We therefore find no evidence of binding of the reported CRD to Gal/GalNAc-containing ligands.

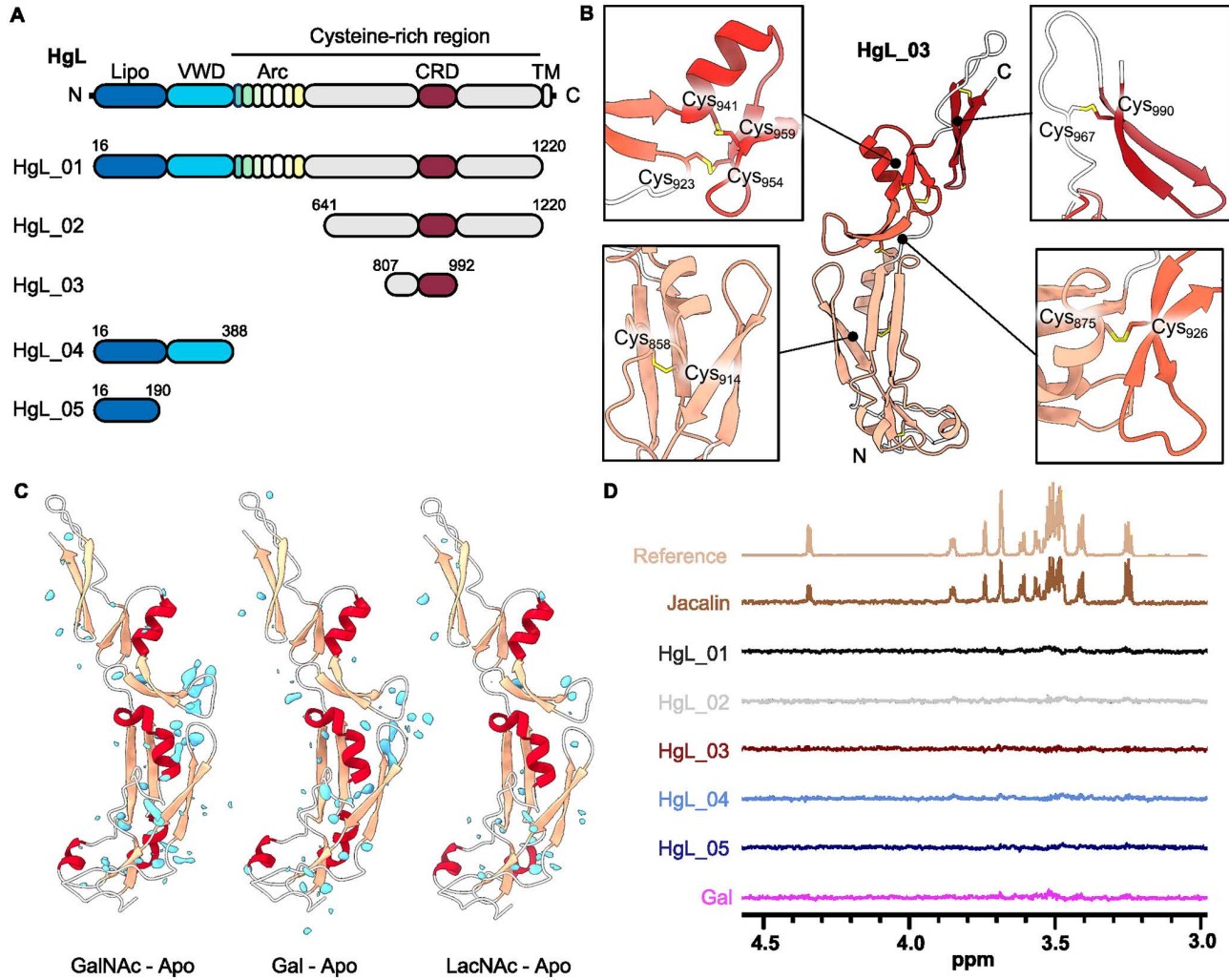

**Fig 1. The previously reported carbohydrate binding domain of the lectin does not bind to Gal/GalNAc-containing carbohydrates. (A)** The domain architecture of HgL and the domain boundaries of expressed constructs. **(B)** Close up of the HgL_03 structure (residues 808-992), highlighting the four modules as insets and showing the locations of the disulphide bonds in yellow. **(C)** Electron density difference map for HgL_03 in the presence of Gal, GalNAc or LacNAc after subtraction of the density in the absence of Gal, GalNAc or LacNAc, respectively, contoured at 2.0. Positive difference density is coloured pale blue. **(D)** STD-NMR showing no evidence of binding of different fragments of HgL to Gal. The scale for the reference spectrum has been adjusted 128-fold relative to those of the lower intensity STD NMR spectra. Jacalin is included as a positive control and no binding is observed to HgL_03. The scale for the reference spectrum has been adjusted 128-fold relative to those of the lower intensity STD NMR spectra.

## Structure of the ordered core of the native dimeric Gal/GalNAc lectin

With no Gal/GalNAc binding observed to recombinantly expressed HgL or fragments, we next switched to study the native protein. The full HgL-LgL heterodimeric complex was purified from axenic cultures of *Entamoeba histolytica* trophozoites by affinity chromatography using monoclonal antibody CP33-H/L-LA 222[2] and its structure was investigated by cryo-EM, yielding reconstructions of the Gal/GalNAc lectin in different states at resolutions ranging from 3.0 Å to 3.9 Å (S1-S3 Figs and S2-S4 Tables). These shared a static globular core consisting of LgL bound to the N-terminal half of HgL (residues 16–584) (Fig 2A-2C), which was sufficiently well resolved to allow model building.

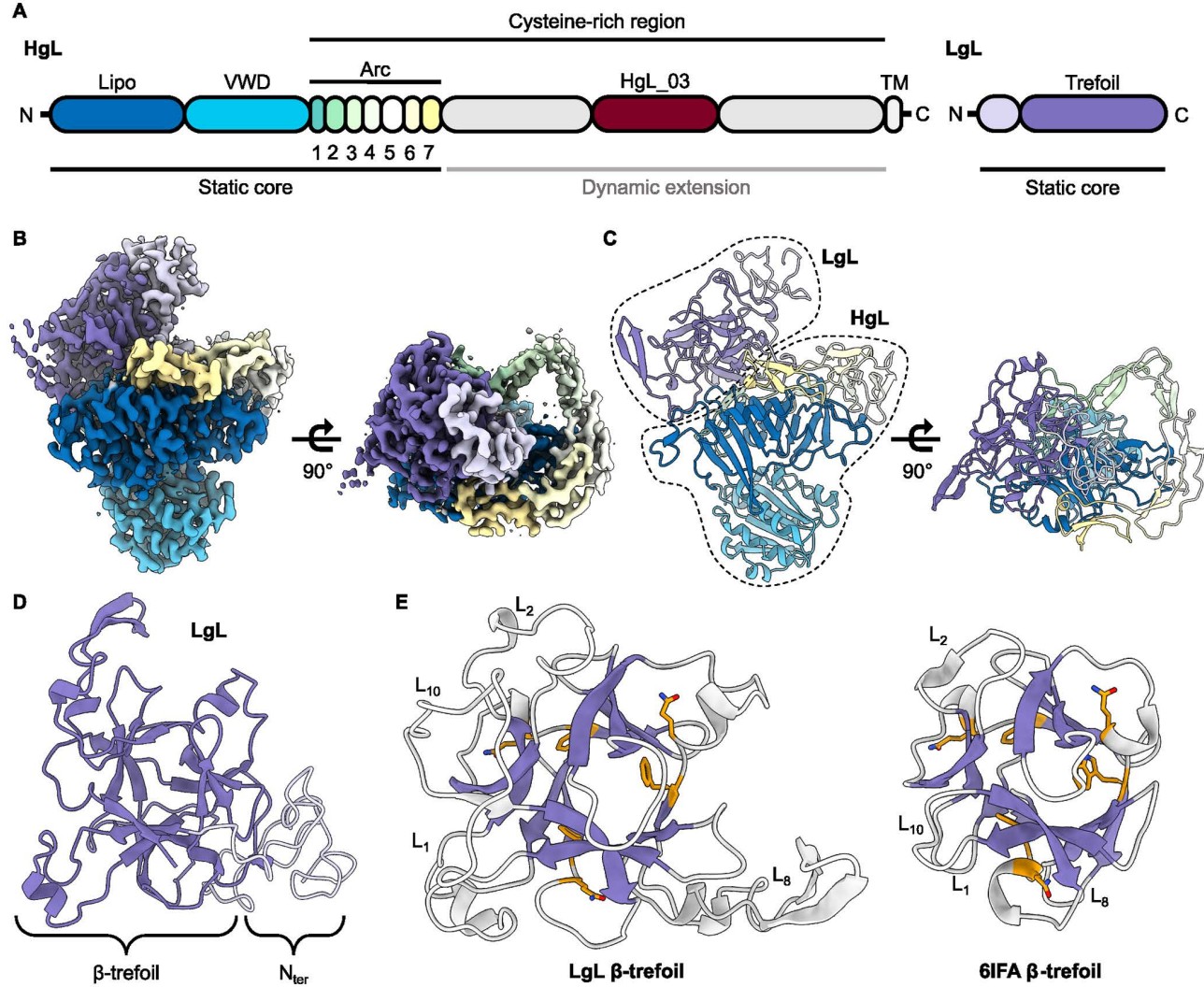

**Fig 2. Structure of the core of the Gal/GalNAc lectin. (A)** A schematic showing domain boundaries for the HgL and LgL chains, derived from the structure. **(B)** Cryo-EM-derived volume for the lectin complex, coloured as **(A)**. **(C)** Cartoon representation of the static core of the lectin, with dashed libes used to delineate the light and heavy chains. **(D)** Ribbon representation of LgL. **(E)** Comparison of LgL (left) and another β-trefoil protein from Entamoeba histolytica (right, PDB code 6IFA) after structural alignment. The characteristic β-strands and the QxF/QxW motifs are highlighted in purple and orange, respectively.

At the centre of the core of the heavy chain lies its N-terminal domain (residues 16–210), a fold formed by a series of eleven antiparallel β-strands and stabilised by six disulphide bonds (Figs 2A-2C and S4A). The two most exposed β-strands ($\beta_6$ and $\beta_7$) are joined by a long loop containing hydrophobic residues which point toward the inner core of the barrel. At the opposite side, a C-terminalα-helix surrounds the N-terminal half ($\beta_1$ to $\beta_6$) of the barrel. A similar fold is found in bacterial lipoprotein carrier proteins such as LolA and LolB. These contain a hydrophobic pocket inside the β-barrel that aids translocation of lipoproteins and complex lipids from the periplasm to the outer membrane. While a similar hydrophobic cavity is found in the Gal/GalNAc lectin, we found no density for a hydrophobic molecule in the cryo-EM reconstructions (S4A Fig), and the barrel appears to be capped at both ends. This domain lies at the centre of the lectin, making multiple interactions with the rest of the globular core.

The lipoprotein-like domain is followed by a peripheral von Willebrand-like domain (VWD, residues 211–392), which consists of a β-sheet of six β-strands surrounded by six α-helices (Figs 2A-2C and S4B). VWD are found in various plasma proteins including integrins, complement factors, collagens and von Willebrand factors, where they mediate interactions with glycoproteins and ligands of the extracellular matrix such as collagen. Ligand binding most often requires cation coordination at a metal ion-dependent adhesion site (MIDAS) formed from conserved residues DXSXS or DXTXS. In the case of the Gal/GalNAc lectin, the VWD lacks a MIDAS (S4B Fig) but may still interact with proteins or ligands in a metal-independent manner. The interface between the VWD and its adjacent domains is stabilised by a $Man_5GlcNAc_2$ N-linked glycan (linked to Asn390, S4C Fig) [24].

The rest of the ectodomain of HgL is a cysteine-rich region containing around 800 residues, of which around 200 residues (393–584) are found in the ordered core. These form an arc that wraps around and contacts the lipoprotein-like domain (Figs 2A-2C and S4D-S4G). The arc consists of seven interconnected pseudo-domains, six of which share a novel fold with a conserved topology, where approximately 20–35 residues form a β-sheet consisting of three anti-parallel β-strands. Each pseudo-domain contains four cysteines, the first and last of which form an intra-domain disulphide bond joining the first and last strands of the domain, while the second and third cysteines form inter-disulphide bonds with the N-terminally and C-terminally adjacent pseudo-domains, respectively.

At the opposite end from the VWD, is the light chain, LgL. This is formed predominantly from a β-trefoil fold of around 200 residues, which is preceded by a sequence without secondary structure (Fig 2D). LgL is covalently bound to HgL through a single disulphide bond between Cys75 of LgL and Cys415 of HgL, and forms extensive hydrophobic contacts with the N-terminal domain and the cysteine-rich arc of HgL (S5 Fig). All C-terminal residues of LgL were resolved in the structure, showing that LgL is not GPI-anchored to the membrane, as previously suggested. Comparison of the β-trefoil of LgL with a previously reported β-trefoil lectin from *Entamoeba histolytica* [25] shows a similar arrangement of β-strands, with most changes occurring in the length of connecting loops (Fig 2E). Indeed, LgL loops are much longer, especially between β-strands $β_1$ and $β_2$ ($L_1$), $β_2$ and $β_3$ ($L_2$), $β_8$ and $β_9$ ($L_8$), $β_{10}$ and $β_{11}$ ($L_{10}$). Notably, LgL contains three QXF sequence motifs commonly found in ricin-type β-trefoil lectins, a family of proteins known to frequently mediate binding to Gal/GalNAc-containing sugars (Fig 2E).

## The light chain contains a Gal/GalNAc binding site

As STD NMR did not reveal binding of Gal/GalNAc to the heavy chain, HgL, we next used the technique to study binding of Gal to the native HgL-LgL heterodimer (Fig 3A). While we again observed no binding to HgL ectodomain, clear binding was seen to the native heterodimer. This is despite our recombinant HgL construct adopting the same global conformation as that seen in the heterodimer when studied by cryo-EM (Fig 3B). This suggested that the formation of the Gal/GalNAc binding site requires the presence of the LgL subunit.

We next collected cryo-EM data for the native HgL-LgL heterodimer incubated with either 20 mM Gal or 20 mM Lac-NAc. We did not observe any major difference in electron density in proximity to HgL when comparing carbohydrate-containing samples with those lacking carbohydrates. In contrast, we observed a single extra density in LgL in the presence of either Gal and LacNAc (Fig 3C and 3D). In Gal and LacNAc-soaked lectin, this density fitted mono- or disaccharide, respectively, with a clear and distinctive difference in density between apo, monosaccharide and disaccharide (Fig 3D). This identified a binding site located in a positively-charged cavity (S6A Fig). Notably, the lactose binding site in the crystal structure of a β-trefoil lectin from earthworm [26] is in an equivalent location (S6B Fig) and high-resolution crystal structures of β-trefoils co-crystallised with lactose were used to guide fitting of sugar models into the LgL electron density (Fig 3D). The binding pocket consists of residues conserved in other CRDs of ricin-type β-trefoils, with Asp159, Asn187, Asn176, Gly162 and Glu123 forming hydrogen bonds to the O2, O3, O4 and O6 atoms of the galactose unit. In addition, important van der Waals interactions are made, such as the stacking interaction between the plane of the indole of Trp173 and the plane made by C3, C4, C5 and C6 of Gal. While some ricin-type β-trefoil lectins bind to Gal/

 

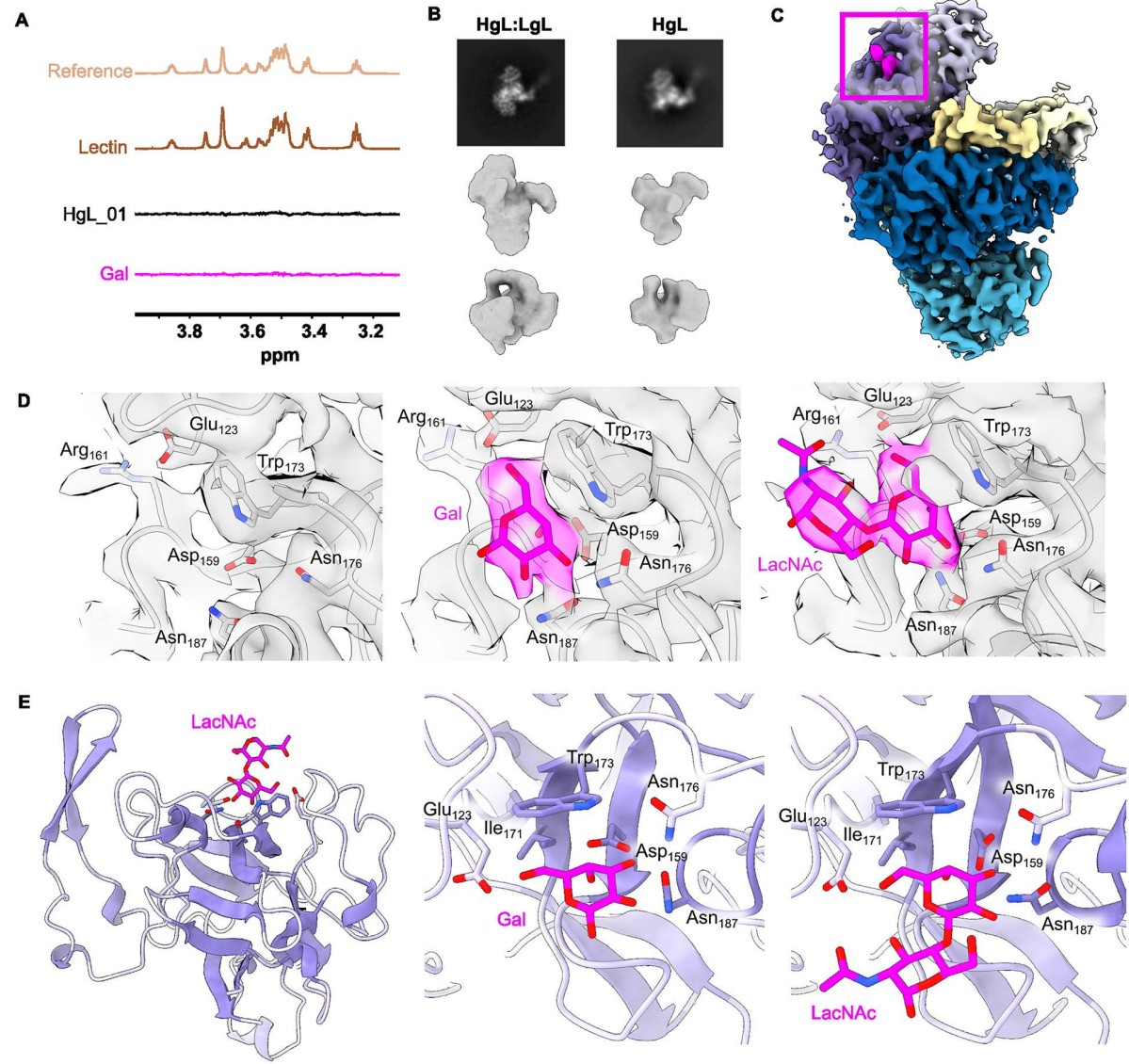

**Fig 3. LgL contains a Gal-binding domain. (A)** Saturation transfer NMR studies of the binding of galactose (Gal) to native HgL-LgL heterodimer (Lectin) as well as to the heavy chain ectodomain. The scale for the reference spectrum has been adjusted 16-fold relative to those of the lower intensity STD NMR spectra. **(B)** Comparison of the native HgL-LgL dimer (left) with recombinant HgL (right) as visualised using cryo-EM. The top panels show 2D class averages while the central and lower panels show two equivalent views of 3D volumes. **(C)** Cryo-EM-derived volume for the LacNAc-soaked lectin. This is coloured as in Fig 2, with LacNAc coloured pink and its location highlighted with a pink box. **(D)** Electron densities focused onto the Gal/GalNAc-binding site of LgL for apo (left), Gal-soaked (central) and LacNAc-soaked (right) native lectin dimer. Coulomb densities are 0.46 for apo, 0.47 for LacNAc-soaked and 0.9 for Gal-soaked. **(E)** Models showing close-ups on the carbohydrate binding pocket and the residues involved in interactions with Gal (central) and LacNAc (right and left).

GalNAc-containing sugars with up to three similar binding pockets, only one such binding pocket is observed in LgL (S6D and S6E Fig). There, the conformation of the carbohydrate-binding pocket in the sugar-bound and apo structures differ in residues 174–190 (S6C Fig). Indeed, upon addition of Gal or LacNAc, the movement of this stretch of residues allows interaction between the carbohydrate and Asn176. Together, our STD NMR and cryo-EM studies therefore reveal no binding to HgL, but instead identify and characterise a single Gal/GalNAc binding site located in LgL.

## The elongated C-terminal arm of HgL adopts multiple conformations

While the ordered core of the lectin remains static, we did not observe clear density for the remainder of HgL. We therefore separated the data into multiple three-dimensional classes. These shared a largely unaltered ordered lectin core, but the HgL C-terminal arm was found to adopt three very different conformations, modes 1–3 (Figs 4A-4C and S7). In modes 1 and 2, we could observe three additional domains, while mode 3 allowed us to visualise a further six domains, together with the region present in HgL_03. In each case these additional domains form a lever arm which projects in different directions relative to the ordered core. Its location in the three modes can be related by rotational movement. The transmembrane helix lies at the end of this arm and is not visualised.

The newly resolved domains mostly adopt similar structures to the seven domains which form the arc. Indeed, most of these domains have the same disulphide bonding pattern as that observed in arc domains, with the first and forth cysteines forming an intradomain disulphide bond while the second and third cysteines form disulphide bond to the previous and subsequent domains, rigidifying the arrangement (S8 Fig). The exceptions are domains 9 and 10, which only contain three cysteine residues. Here, the first and third cysteine residues form the intradomain disulphide bond, leaving only one cysteine free for interchain disulphide formation. This results in a lack of a disulphide bond linking domains 9 and 10, suggesting the potential for increased flexibility in this part of the arm. Additionally, a large conformational change in the linker between the last domain of the arc (pseudo-domain 7) and the first domain of the level arm is observed. This forms a major pivot point and causes a ~90° change in the direction of the arm, allowing substantial rotation as the arm transitions between the locations observed in modes 1 and 3 (Fig 4D).

## Discussion

Like many enteric pathogens, *Entamoeba histolytica* must navigate through a protective mucus layer before reaching the intestinal epithelium. The single cell amoeba binds to Gal and GalNAc residues of colonic mucins using its Gal/GalNAc surface lectin [10]. After degradation of mucin glycoproteins by secreted cysteine proteases [27], the same protein complex mediates attachment to exposed host epithelial cells, a prerequisite for cytolysis and invasion of these cells [9,28]. The Gal/GalNac lectin is therefore important for pathogenesis and forms an important vaccine candidate.

Previous reports suggested that a small stretch of residues in the cysteine-rich portion of HgL mediates carbohydrate binding [14]. However, we could not replicate these findings using various recombinant constructs that encompass the proposed carbohydrate binding domain. We find no structural similarity between this domain and any known lectin and find no additional density corresponding to sugar after its co-crystallization with Gal/GalNAc-containing sugars. Finally, when we study carbohydrate binding in physiological buffer conditions, we observe no binding to the proposed Gal/GalNAc binding domain, or indeed to any part of HgL, despite our observation of binding to native lectin purified directly from *Entamoeba histolytica*. Our data therefore cast doubt on the presence of a Gal/GalNAc binding site in the lectin heavy chain.

In contrast, our structural studies of native HgL-LgL heterodimer revealed that the light chain, LgL, contains a single carbohydrate binding site. Indeed, LgL consists predominantly of a β-trefoil domain, which is a known lectin fold [29]. The functional importance of LgL has been largely overlooked to date, most likely due to the difficulty of purifying it in isolation. Instead, most of our understanding of the light chain function comes from deletion studies performed in parasites without prior understanding of its structure, or from the observation that monoclonal antibodies raised against fusion-tagged LgL do not inhibit adherence [30], although these do not bind to whole fixed trophozoites, suggesting incorrect folding of the material used to raise antibodies [31]. We also found that we could not purify LgL recombinantly in various expression systems, while we could express correctly folded HgL fragments from a mammalian recombinant system. Further studies to purify monomeric LgL, guided by our structures, will allow further characterization of carbohydrate binding and will also enable production of specific antibodies that may directly prevent binding to mucins.

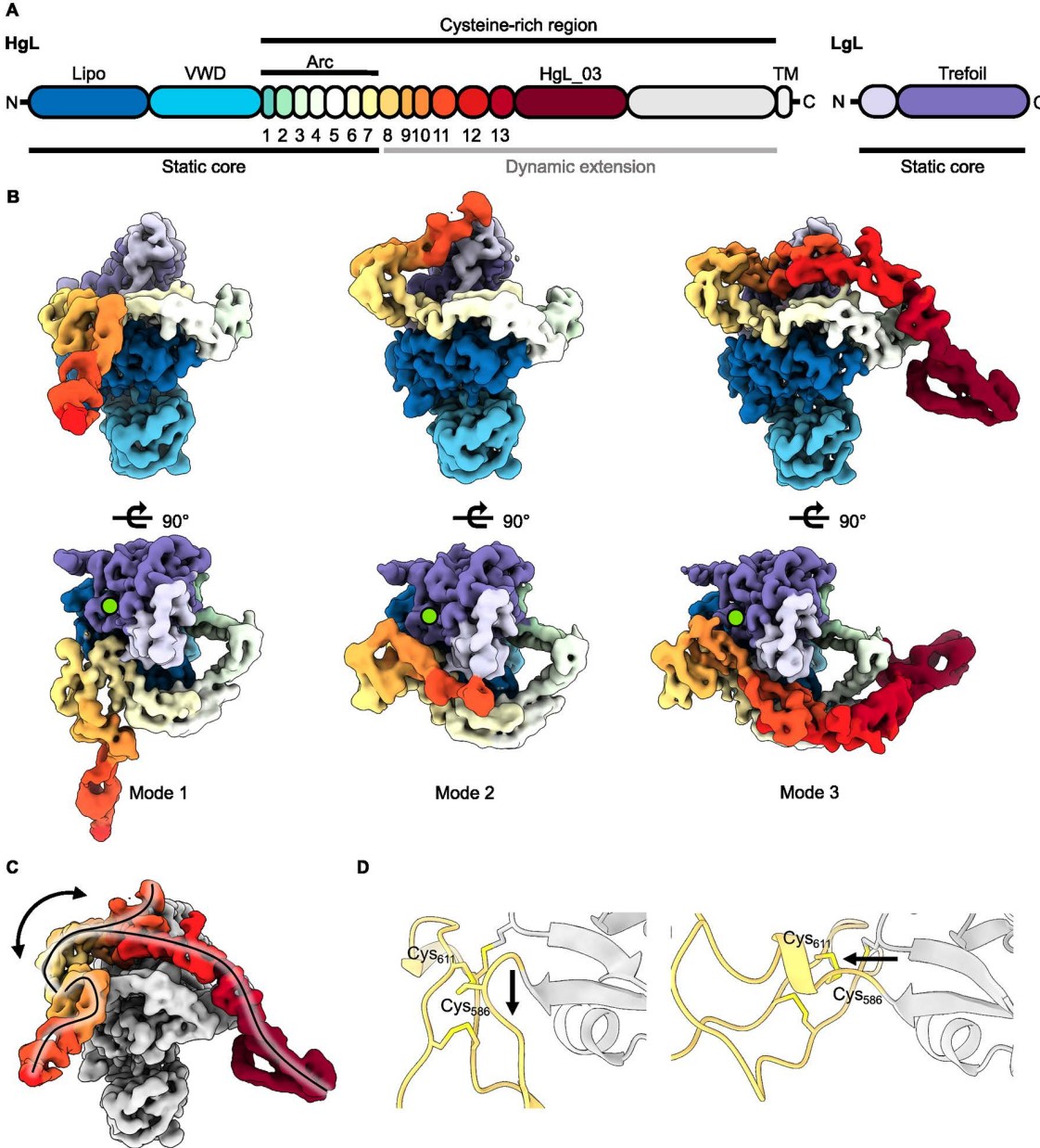

**Fig 4. The C-terminal part of HgL forms a dynamic arm. (A)** Domain boundaries for the HgL and LgL chains, derived from the structures obtained after 3D classification of the C-terminal half of HgL. **(B)** Three different conformations of the dynamic extension of the lectin, revealed after 3D classification of cryo-EM data. The colours are as in **(A)**. The galactose binding site is labelled with a lime green circle. **(C)** Overlay of the three conformations. The paths taken by the arm are highlighted by the lines, with a two-sided arrow indicating the likely motion to transition between conformations. The globular core is coloured in grey, the arm is coloured as in **(A)**. **(D)** A close-up of the structures of mode 1 (left) and mode 3 (right), focusing on the junction between arc domain 7 and domain 8. The black arrows highlight the two different directions followed by the polypeptide backbone at this hinge-point.

While our studies resolve the function of the light chain, LgL, they raise some questions about the role of the heavy chain, HgL, which does not bind to Gal/GalNAc-containing sugars in our STD NMR experiments. Our structure reveals that HgL consists of a substantial ordered core attached to an elongated arm. The ordered core contains domains which

share a fold with a lipoprotein-like domain and a von Willebrand-like domain. While these structural similarities might indicate functions such as lipid or extracellular matrix binding, no binding partner has yet been identified. Also intriguing is the elongated arm, which is rigidified by interdomain disulphide bonds, but which can pivot around the joint between the arm and the globular core. Interestingly, in one of the observed conformations (mode 3), the lever arm adopts a position in which it lies very close to the carbohydrate binding site. This suggests that the presence of bulky Gal/GalNAc-containing glycoproteins, such as mucins is likely to favour the alternative conformations. Future studies aimed at deciphering the interplay between carbohydrate binding and lever arm dynamics will likely reveal the function of this lever arm.

## Methods

### Protein expression and purification

Synthetic genes encoding full-length HgL (isoform EHI_133900) and the reported variable regions of antibody CP33-H/L-LA 222[2] were codon-optimised for expression in mammalian cells (Invitrogen). Genomic fragments were amplified by PCR using PCR mastermix 2X (#K0171, ThermoFisher Scientific) or KOD Xtreme Hot Start DNA polymerase kit (#71975, Novagen) for fragments exceeding 3 kilobase pairs in size. For HgL, sequences were inserted by Gibson Assembly (#E5510, New England BioLabs) into a pHLSec vector [32] containing an N-terminal secretion signal and a 6xHis C-terminal tag. For Fab production, the sequences of the variable regions CP33-H and L-LA22 were inserted into pOPINVH and pOPINVL [33], respectively. DNA sequencing reactions were performed by Source Bioscience.

Recombinant proteins were transiently expressed in Expi293F cells cultured in suspension using the Expi293 expression system kit (ThermoFisher Scientific) in the presence of the 1 µM kifunensine. For Fab production, pOPINVH-CP33 and pOPINVL-LA22 were co-expressed. After five days, culture supernatants were harvested by centrifugation, filtered, and incubated with Ni Sepharose excel resin (Cytiva). Beads were washed with 25 mM Tris pH 8.0, 300 mM NaCl, 10 mM imidazole and bound proteins were eluted with 25 mM Tris pH 8.0, 300 mM NaCl, 300 mM imidazole. For deglycosylation, Endonuclease Hf (#P0703, New England BioLabs) was added and incubated overnight at room temperature before application on size-exclusion chromatography column (Superdex S200 or S75 Increase 10/300 GL, Cytiva) in 20 mM HEPES pH 7.4, 150 mM NaCl.

### Axenic culture of *Entamoeba histolytica* and extraction of the Gal/GalNAc lectin

A starter axenic culture of HM1:IMSS trophozoites was kindly provided by Professor Graham Clark (London School of Hygiene and Tropical Medicine, the United Kingdom). Trophozoites were cultured in homemade LYI-S2 axenic medium in a static incubator at 35.5°C. Cultures were scaled up from glass tubes (13 mL) to larger T-flasks. For each passage, adherent parasites were washed with pre-warmed PBS and detached in LYI-S2 medium on ice. Flasks were filled to 90% capacity with LYI-S2 medium, sealed and incubated vertically for up to three days before passage or harvest.

After 72 hours, confluent trophozoites were harvested by centrifugation at 300 *g* for 20 minutes and washed three times with PBS. To extract the Gal/GalNAc lectin, a 5 g pellet of trophozoites was resuspended in PBS supplemented with EDTA-free protease inhibitors. The suspension was Dounce-homogenised, briefly sonicated and solubilised in 1% lauryl maltose neopentyl glycol (LMNG) at 4°C overnight. After centrifugation at 20,000 rpm for 1 h at 4°C, the lysate was filtered and further purified by immuno-affinity chromatography. For this, the Fab portion of antibody CP33-H/L-LA22 was covalently coupled to a cyanogen bromide-activated Sepharose 4B resin (Sigma) at a concentration of 10 mg Fab per mL of resin, as recommended. The lysate was incubated with resin at 4°C overnight and washed with PBS supplemented with LMNG at twice the critical micellar concentration (CMC), eluted with 100 mM glycine pH 2.7, 2 CMC LMNG into 0.5 mL fractions containing 50 µL of 1 M Tris pH 9.0 for neutralisation.

After affinity chromatography, the fractions containing the proteins were pooled together and concentrated with a 100K Amicon Ultra centrifugal unit (Millipore) before application on size-exclusion chromatography column (Superdex S200

10/300 GL, Cytiva) in 20 mM HEPES pH 7.4, 100 mM NaCl, 1.2 CMC LMNG. The fractions were analysed by SDS-PAGE in reducing and non-reducing conditions.

## Crystallographic studies of HgL_03

All proteins were deglycosylated and applied to size-exclusion chromatography before crystallization, as described above. Crystallization screens were set up in a 1:1 volume ratio at a protein concentration of 60 mg/mL for HgL_03. Crystals of HgL_03 were obtained within three days at 4°C in 0.05 M Sodium acetate, pH 4.0, 0.225 M Ammonium sulfate, 12% w/v PEG 4000 in the absence of carbohydrate or in the presence of either 500 mM Gal, 500 mM GalNAc, or 50 mM LacNAc. Crystals were cryo-protected using crystallization solutions supplemented with 20% glycerol and cryo-cooled into liquid nitrogen.

X-ray diffraction data derived from crystals of HgL_03 in the absence and in the presence of carbohydrates were collected on beamline I04 (Diamond Light Source, UK). Diffraction data for HgL_03 in the presence of GalNAc and LacNAc were auto-processed in xia2-dials [34,35] to 1.85 Å and 2.32 Å, respectively. Data for HgL_03 alone or in the presence of Gal were auto-processed in autoPROC and staraniso [36] to 2.48 Å, 1.82 Å and 3.03 Å, respectively. Molecular replacement for HgL_03 was performed with PHASER [37] using a model generated by an Alphafold2 v2.1 [22] prediction of residues 820–920 of HgL as search templates. Side chains were truncated to alanine and flexible loops were deleted before molecular replacement. Model building and refinement were performed iteratively in COOT [38], phenix.refine [39] and BUSTER [40].

## Saturation transfer difference by nuclear magnetic resonance

Prior to the experiments, protein samples were buffer-exchanged into 10 mM deuterated Tris pH 7.5, 100 mM NaCl, 10 mM $CaCl_2$, 1 mM $MgCl_2$ using Amicon Ultra centrifugal units (Millipore) washed with $D_2O$. 1 M stock solutions of carbohydrates were prepared in $D_2O$. Sample volumes of 150–200 µL were loaded in 3 mm outer diameter SampleJet NMR tubes (Bruker). The folded state of protein samples was verified by collecting 1D 1H NMR spectra for all samples. A protein concentration of 10 µM was used with a ligand molar excess of 500.

All NMR experiments were conducted on a 950 MHz spectrometer equipped with an Oxford Instruments magnet, a Bruker Avance III HD console, a four channel 5 mm 1H-optimised triple resonance helium-cooled cryoprobe and a SampleJet sample changer (Bruker). STD NMR experiments were performed at 10°C using a pulse sequence described previously [41] and an excitation sculpting water-suppression scheme [42]. Protein signals were suppressed in STD NMR using a 30 ms spin-lock pulse.

Data were collected for a sweep width of 10.97 ppm (dwell time of 48 µs) and 32,768 complex points. On- and off-resonance irradiation data were separated into 16 blocks of 8 transients each (128 total transients per irradiation frequency), using a recycle delay of 4 s and saturation for 3.5 s at 7.5 ppm (aromatic residues) or 26 ppm, respectively. For each experiment, the total acquisition time was approximately 30 min.

Data were processed using TopSpin 3.6.1 (Bruker). Reconstructed time-domain data from the difference of on- and off-resonance irradiation (STD spectra) or only the off-resonance irradiation (reference spectra) were processed by applying a 2 Hz exponential line broadening function and zero-filling to 65,536 points prior to Fourier transformation. Phasing parameters were derived for each sample from the reference spectra and copied to the STD spectra. 1H peak intensities were integrated in TopSpin 3.6.1 (Bruker) after baseline correction.

## Cryo-EM studies of the Gal/GalNAc lectin

In a first experiment, the protein purified by size-exclusion chromatography was either applied directly onto grids or pre-incubated with 20 mM LacNAc for 30 min. In a second experiment, the protein purified from a different preparation was first incubated with 20 mM Gal for 30 min, then PEGylated *in situ* using the methyl-PEG-NHS-ester reagent MS(PEG)$_{12}$

(#22685, ThermoFisher Scientific). PEGylation was performed to reduce preferred orientation, with a 40-fold molar excess of PEG reagent used to PEGylate an estimated one third of the total lysines. The reaction was incubated for 2 hours on ice and quenched by addition of 50 mM Tris pH 7.5, 100 mM NaCl, 1.2 CMC LMNG before application onto grids.

Cryo-EM grids were prepared with a FEI Vitrobot Mark IV (ThermoFisher Scientific) at 4°C and 100% humidity. 3.5 µL of protein sample at 0.15 mg/mL for the Gal/GalNAc lectin were applied onto glow-discharged grids (Quantifoil R2/1 carbon film on a 300 gold mesh). After incubation for 10 s, grids were blotted with a blot time of 1–2 seconds before plunge-freezing into liquid ethane.

Cryo-EM grids were imaged using a Titan Krios G3 (ThermoFisher Scientific) transmission electron microscope operating at 300 kV and equipped with a BioQuantum imaging filter (Gatan). Data were recorded on a K3 direct electron detector (Gatan) under low dose conditions with a nominal pixel size of 0.832 Å per pixel. Details of the data collection strategy are given in S2-S4 Tables.

**Cryo-EM data processing**

Motion correction of all movies was performed in SIMPLE 3.0 [43]. For each dataset, a subset of 500 movies were selected for contrast transfer function (CTF) determination, particle picking and 2D classification in SIMPLE 3.0, yielding representative 2D references. All micrographs were imported to cryoSPARC v3.3 [44], where all further processing steps were carried out. After CTF estimation, exposures were curated by CTF fitting and ice thickness. A total of 24,194 (lectin), 23,231 (lectin + LacNAc) and 11,996 (lectin + Gal) exposures were selected. Particles were picked using Template picker and the 2D references generated in SIMPLE 3.0, yielding a total of 8,055,928 (lectin), 7,594,780 (lectin + LacNAc) and 6,391,082 (lectin + Gal) particles. After particle extraction with a box size of 256 x 256 pixels, each dataset was randomly split into stacks of 4,000–5,000 movies. For each stack, 2D classification was performed with a total of 500 classes and 40 iterations, yielding a total of 1,548,634 (lectin), 2,189,906 (lectin + LacNAc) and 1,359,171 (lectin + Gal) particles after manual selection. The rarest 2D views and their corresponding particles were used as inputs for picking in Topaz with 20 epochs and 100-to-250 particles per movie. Three iterative rounds of training were performed on 500 movies per dataset. Using these models, particle picking was performed for entire datasets, followed by extraction with a box size of 256 x 256 pixels and 2D classification, leading to the selection of 960,785 (lectin), 1,344,401 (lectin + LacNAc) and 1,260,323 (lectin + Gal). Ultimately, particles obtained from Template picker and Topaz jobs were combined and duplicates were removed, resulting in a total of 1,890,286 (lectin), 2,826,892 (lectin + LacNAc) and 1,795,334 (lectin + Gal) particles.

For each dataset, particles were used for 3D *ab-initio* reconstruction into three classes. Two classes were selected for heterogenous refinement into three classes using both *ab-initio* reconstructions and a decoy volume, consistently yielding a volume with good angular distribution corresponding to the lectin core, a similar volume with a lever arm characteristic of mode 1 characterized by pronounced preferred orientation and streaky artefacts, and a poorly resolved volume. A total of 773,328 (lectin), 1,068,080 (lectin + LacNAc) and 903,803 (lectin + Gal) particles representative of the first volume were selected for non-uniform refinement, yielding maps of the globular core of the lectin at a global resolution of 3.4 Å (lectin), 3.0 Å (lectin + LacNAc) and 3.0 Å (lectin + Gal). To distinguish the different conformations adopted by HgL lever arm, particles were subjected to 3D classification into 10 classes.

For the dataset of the lectin without carbohydrate, 556,899 particles (mode 2) were selected and subjected to non-uniform refinement, yielding a final reconstruction at a global resolution of 3.5 Å (2.9 to 5.3 Å range). For modes 1 and 3, 112,643 particles and 103,786 particles were selected, respectively, yielding reconstructions at a global resolution of 3.7 Å after non-uniform refinement. To better resolve the extension, 3D classification into five classes was performed. 98,775 particles (mode 1) and 47,934 particles (mode 3) were selected and subjected to non-uniform refinement, yielding final reconstructions at a global resolution of 3.7 Å (3.1 to 6.6 Å range) and 3.9 Å (3.3 to 8.1 Å range), respectively.

In the presence of LacNAc, 222,411 particles (mode 1) were selected and subject to non-uniform refinement and CTF refinement, yielding a final reconstruction at a global resolution of 3.3 Å (2.8 to 6.8 Å range). Further heterogenous

 

refinement of mode 2 and mode 3 particles into four classes was performed using their respective reconstructions obtained from the untreated dataset and two decoy volumes as 3D references, leading to selection of 125,174 particles (mode 2) and 60,909 particles (mode 3), which yielded final reconstructions at a global resolution of 3.4 Å (2.9 to 6.6 Å range) and 3.9 Å (3.3 to 10.1 Å range) after non-uniform refinement, respectively. Here, we note that the volume representative of mode 3 is of relatively poor quality, specifically at the lever arm, suggesting that only a minor population of particles adopts this conformation.

In the presence of Gal, 445,091 particles (mode 1) and 94,997 particles (mode 2) were selected for non-uniform refinement and CTF refinement, yielding reconstructions at a final global resolution of 3.1 Å (2.6 to 6.3 Å range) and 3.4 Å (2.9 to 7.2 Å range), respectively. For mode 3, 100,529 particles were selected, yielding a reconstruction at a global resolution of 3.2 Å after non-uniform refinement. To better resolve the extension, these were further classified into ten classes. A subset of 10,939 particles was selected for non-uniform refinement, yielding a final reconstruction at a global resolution of 3.9 Å (3.5 to 8.4 Å range).

### Model building and refinement

Model building of the lectin heterodimer was guided by the models of the β-trefoil of LgL (EHI_035690, residues 75–288) and the von Willebrand-like domain of HgL (EHI_133900, residues 211–392) predicted with high confidence by Alphafold2 v2.1 [22]. Starting models were fitted into the cryo-EM map of the lectin in mode 1 in ChimeraX [45] and used as a starting reference for *de novo* model building of the rest of the globular core in COOT [38]. The resulting model of HgL:LgL core was refined using phenix.real_space_refine [39] with global minimization and secondary structure restraints against the postprocessed cryo-EM map of the lectin in mode 1. Using this starting model, HgL lever arm was built into the final maps of the three modes. Model building was guided by fitting subdomains encompassing residues 654–669, 692–730 and 774–802 of HgL derived from its prediction in Alphafold2 v2.1. *De novo* building was performed for the rest of HgL arm until residue 670 for mode 1, residue 669 for mode 2 and residue 819 for mode 3, after which density was poorly resolved. Each structure was refined using phenix.real_space_refine [39] with global minimization and secondary structure restraints against their corresponding postprocessed cryo-EM maps. Details of the refinement and validation statistics are given in S2-S4 Tables.

Structural homologues of HgL lipoprotein-like domain, von Willebrand-like domain and LgL β-trefoil were identified by DALI [46] using the PDB25 database. These were used to guide docking of Gal and LacNAc in the carbohydrate binding pocket of LgL after structure alignment with different β-trefoils in complex with lactose (RCSB codes 5Y97, 4IZX, 3NBD, 2ZQN and 1PUU).

### Supporting information

**S1 Fig. Cryo-EM processing scheme for lectin.** The scale bar corresponds to 50 nm.
(TIF)

**S2 Fig. Cryo-EM processing scheme for lectin in the presence of Gal.** The scale bar corresponds to 50 nm.
(TIF)

**S3 Fig. Cryo-EM processing scheme for lectin in the presence of LacNAc.** The scale bar corresponds to 50 nm.
(TIF)

**S4 Fig. Structures of the domains of HgL globular core.** (A) Rainbow representation of the lipoprotein-like domain of HgL (Lipo). The rectangle highlights the site which binds lipid-like molecules in homologous structures. Below is a close-up showing the electron density in the cavity and at the centre of the lipoprotein-like domain, revealing the absence of density due to a ligand. (B) Rainbow representation of the von Willebrand-like domain of HgL (VWD). The location of a potential

MIDAS is delimited by a rectangle. A close-up showing the structural alignment of the VWD (rainbow) and the MIDAS of $\alpha_1$-$\beta_1$ integrin (PDB: 1PT6, grey) is displayed below. (C) Cartoon representation of the Lipoprotein-like domain (Lipo, dark blue) and the von Willebrand-like domain (VWD, light blue), overlaid with the electron density of the N-glycan linked to Asn390 (brown) stabilising the interface between both domains. Close up on the $Man_5GlcNAc_2$ glycan at the interface between the lipoprotein-like domain and the von Willebrand-like domain. (D) Sequence of each pseudo-domain, with cysteines coloured. Bridges between cysteines represent disulphide bonds. (E) Cartoon representation of the arc of pseudo-domains with disulphide bonds coloured as (A). (F) A close-up on a portion of the arc of pseudo-domains showing the interconnection between adjacent pseudo-domains. (G) Schematic representation of the topology adopted by the pseudo-domains.
(TIF)

**S5 Fig. The interactions between LgL and HgL.** (A) The single intramolecular disulphide bond links HgL and LgL. (B) Surface hydrophobicity of LgL, showing the interface between LgL and HgL (left) and solvent exposed side of LgL (right).
(TIF)

**S6 Fig. Structural analysis of carbohydrate binding by LgL β-trefoil.** (A) Surface electrostatic representation of galactose-bound LgL. (B) Cartoon representation of the alignment between the structural models derived from the cryo-EM reconstruction of carbohydrate-bound HgL-LgL heterodimer (purple) and a lactose-bound beta trefoil (light grey, PDB code 2ZQN) showing an analogous carbohydrate binding pocket. (C) Cartoon representation of the alignment between the structural models derived from the cryo-EM reconstruction of HgL-LgL heterodimer bound to Gal (orange) and LacNAc (purple), and in the absence of carbohydrate (light grey). (D) Sequence of the three pseudo-symmetric portions of LgL β-trefoil. Residues forming the twelve β-strands of the β-trefoil are highlighted in grey. The QXF motif is displayed in bold. (E) Close-ups on the cartoon representations of the three potential carbohydrate binding pockets of LgL β-trefoil and their overlay with the corresponding cryo-EM density.
(TIF)

**S7 Fig. Cryo-EM reconstructions (top) of the three main different modes observed, and their corresponding 2D classes (below).**
(TIF)

**S8 Fig. Domain arrangement of HgL arm.** (A) Sequence of each domain of the arm and the N-terminally adjacent pseudo-domain of the arc, with coloured cysteines. Bridges between cysteines represent disulphide bonds. (B) Cartoon representation of the resolved domains of HgL arm with disulphide bonds coloured as (A). (C) Close-ups on the different domains and their interfaces.
(TIF)

**S1 Table. Crystallographic data collection and refinement statistics for HgL_03 alone and co-crystallized with Gal, GalNAc or LacNAc.**
(DOCX)

**S2 Table. Cryo-EM data collection, refinement, and validation statistics for the Gal/GalNAc lectin.**
(DOCX)

**S3 Table. Cryo-EM data collection, refinement, and validation statistics for the Gal/GalNAc lectin in the presence of Gal.**
(DOCX)

**S4 Table. Cryo-EM data collection, refinement, and validation statistics for the Gal/GalNAc lectin In the presence of LacNAc.**
(DOCX)

## Acknowledgments

We thank William Petri for an initial sample of the lectin used for early negative stain analysis and Graham Clarke for providing *Entamoeba histolytica*. We thank Ed Lowe and the beamline scientists at ESRF and Diamond Light Source for support with crystallographic data collection. Cryo-EM data was collected at the COSMIC facility and we thank Rishi Matadeen, Teige Matthews-Parmer and Joe Caesar for support. We thank Hannah Ivison for lab management support.

## Author contributions

**Conceptualization:** Samuel F. Gérard, Matthew K. Higgins.

**Formal analysis:** Samuel F. Gérard, Matthew K. Higgins.

**Funding acquisition:** Matthew K. Higgins.

**Investigation:** Samuel F. Gérard, Christina Redfield, Matthew K. Higgins.

**Methodology:** Samuel F. Gérard, Christina Redfield.

**Supervision:** Christina Redfield, Matthew K. Higgins.

**Validation:** Samuel F. Gérard, Matthew K. Higgins.

**Visualization:** Samuel F. Gérard, Matthew K. Higgins.

**Writing – original draft:** Samuel F. Gérard, Matthew K. Higgins.

**Writing – review & editing:** Samuel F. Gérard, Matthew K. Higgins.

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
