## [Decision Letter · Decision Letter 0]

6 Jan 2026

Structural basis for carbohydrate recognition by the Gal/GalNAc lectin of Entamoeba histolytica

PLOS Pathogens

Dear Dr. Higgins,

Thank you for submitting your manuscript to PLOS Pathogens. As you will see, the reviewers appreciated the structural data on the Entamoeba histolytica adhesin/lectin with a novel organization and a sugar binding site in the light chain rather than in the heavy chain, as had been wrongly reported. Yet, the reviewers have found that the study lacks experimental validation of the residues identified by the nice structures reported. Therefore, we invite you to submit a revised version of the manuscript that addresses as much as it is feasible the points raised during the review process.

We look forward to receiving your revised manuscript.

Kind regards,

Félix A. Rey

Academic Editor

PLOS Pathogens

Dominique Soldati-Favre

Section Editor

Sumita Bhaduri-McIntosh

Editor-in-Chief

PLOS Pathogens

orcid.org/0000-0003-2946-9497

Michael Malim

PLOS Pathogens

orcid.org/0000-0002-7699-2064

**Journal Requirements:**

- ® on pages: 8, and 9

- TM on pages: 7, and 8.

4) We notice that your supplementary Figures, and Tables are included in the manuscript file. Please remove them and upload them with the file type 'Supporting Information'. Please ensure that each Supporting Information file has a legend listed in the manuscript after the references list.

5) In the online submission form, you indicated that your data will be submitted to a repository upon acceptance. We strongly recommend all authors deposit their data before acceptance, as the process can be lengthy and hold up publication timelines. Please note that, though access restrictions are acceptable now, your entire minimal dataset will need to be made freely accessible if your manuscript is accepted for publication. This policy applies to all data except where public deposition would breach compliance with the protocol approved by your research ethics board. If you are unable to adhere to our open data policy, please kindly revise your statement to explain your reasoning and we will seek the editor's input on an exemption.

2) If any authors received a salary from any of your funders, please state which authors and which funders..

7) Please send a completed 'Competing Interests' statement, including any COIs declared by your co-authors. If you have no competing interests to declare, please state "The authors have declared that no competing interests exist". Otherwise please declare all competing interests beginning with the statement "I have read the journal's policy and the authors of this manuscript have the following competing interests"

**Reviewers' Comments:**

Reviewer's Responses to Questions

**Part I - Summary**

Reviewer #1: The work primarily focuses on structural characterization of the extracellular part of a heterodimeric complex of a membrane protein that plays a crucial role in the host interaction of the parasite Entamoeba histolytica. This structure is noteworthy in several aspects. Firstly, a novel structural fold is identified, which is rare enough to be highlighted. Secondly, this work identifies an error in the allocation of a carbohydrate binding site. It was previously believed to be on the larger subunit, but it is actually on the smaller one. The authors present strong evidence for this and identify the new site with the ligand inside. Finally, they have identified an elongated arm with multiple potential conformations that could play a significant role in the accessibility of this carbohydrate binding motif, thereby facilitating the interaction with the host and the initiation of the parasite invasion.

The strength of this work lies in its meticulous and precise structural characterization, achieved through both X-ray, STD NMR and cryoEM approaches. However, the weakness is the limited functional characterization, primarily limited to structural descriptions. The reader may feel a bit frustrated after the detailed and informative descriptions of these structures, without seeing physiologic characterization that aligns with the proposed hypothesis.

However, the work is really well conducted, and these new structures and characterizations are without doubt of great importance in the corresponding field and studies around the invasive mechanisms of this parasite. I have no reason not to recommend the publication of this nice work.

Reviewer #2: The results reported in this manuscript provide evidence that the sugar-binding site in the Entamoeba histolytica adhesin is located in a beta-trefoil domain in the light chain of the adhesin. The structures reported also provide evidence that the overall structure of the adhesin exists in multiple conformations.

Although the findings contradict previous studies suggesting that the sugar-binding site is located in the heavy chain, the combined EM structures and TrNOE NMR data provide compelling evidence that the binding site is located in the light chain. The extensive structural analysis using both crystallography and EM also provides an overview of the organization of the adhesin. Both of these results represent significant achievements and are generally well supported by the data.

Reviewer #3: The manuscript reported the first crystal structure of the carbohydrate binding binding domain (CRD) of HGL (HgL_03) from Gal-GalNAc cell surface lectin, from the enteric parasite Entamoeba histolytica and cryoelectron microscopy structures for the native lectin (HGL-LGL heterodimer) extracted from the amoeba. The authors also solved the structures of the co-crystals grown in the presence of Gal/GalNAc or LacNAc.

The structure of the lectin consisted of a core ordered part made up of N terminal half of HGL and LGL bound to it.

HGL in the hetero-dimer revealed existence of a bacteiral lipoprotein like domain in the N-terminal ordered core followed by a Von Willebrand-like domain. Further C terminal Cysteine-rich region consists of a hinge and followed by a dynamic arm. The arm was shown to adopt three distinct conformations, mode 1-3.

LGL showed beta-trefoil fold and was found to be covalently linked to HGL via disulphide bond.

The lectin is known for its very crucial role in adhesion of the amoeba to the host gut via its carbohydrate binding function and thereby promoting amoebic pahthogenesis. The author studied carbohydrate binding by HgL_03 and showed that it does not show any binding to Galactose or LacNAc. Instead, LgL showed binding to Galactose. Based on the cryo electron microscopy electron density map, the authors modeled Gal and LacNAc binding to LGL.

The results reported in the manuscript certainly leads to an advancement in terms of structural information about one of the most well studied amoebic antigens. Interestingly, it reported some observations which is in sharp contrast to the existing literature. It also revealed the existence of domains with implications in lipid transport possible hinting new functions of the lectin.

However the manuscript may gain more strength with more supporting observations using alternative experimental approach as well as inclusion of some of the results as controls.

The method section may need more attention with elaboration of the text and inclusion of results as supplementary information.

**Part II – Major Issues: Key Experiments Required for Acceptance**

Reviewer #1: nothing to ask for

Reviewer #2: (1) The manuscript is focussed on the experimental analysis and does not provide major biological insights because of a lack of links between the structural features observed and the biological functions of the adhesin. Thus, the manuscript does not present the work in a very accessible or compelling way and needs significant refocus, as discussed in the next section.

(2) Analysis in vitro and in vivo of the effects of mutations in the sugar-binding site and the proposed hinge region would substantiate the conclusions and provide at least some linkage to biological functions.

Reviewer #3: 1. Authors could use Gal-GalNAc instead of LacNAc in all carbohydrate binding experiments, as this di-saccharide showed the most effective binding to the CRD as compared to Gal or GalNAc in the earlier studies, reflected from multiple experimental approaches including cell adhesion/agglutination experiments, ITC,etc.

2. A concentration dependent binding experiment could be performed to determine affinity for the carbohydrates by the native hetero-dimer.

**Part III – Minor Issues: Editorial and Data Presentation Modifications**

Reviewer #1: 1) Identify N and C-terminal motifs of the protein in Figure 1.

2) The authors mention in the text the existence of 3 QXF sequence motifs in Lgl which traditionally mediated binding to Gal/GalNAc containing sugars. However, they do not pinpoint those in Figure 4 where the carbohydrate binding mode is presented. It creates some confusion and needs to be clarified in one way or another on the real role of these motifs in the particular case of this protein.

Reviewer #2: (1) Neither the title or abstract highlight key findings for potential readers. For instance, the title could refer to sugar-binding "adhesin" to reflect its biological significance and the abstract makes no mention that the sugar-binding domain falls into a known class of protein folds that support this activity.

(2) No view of the overall organization with respect to the membrane is shown and it is not indicated if the two subunits are encoded in separate genes or are produced by proteolysis.

(3) A great deal of the manuscript is given over to descriptions of negative results. Roughly the first third of the Results section, including the one of the four figures are all negative results before anything positive is reported.

(4) In addition to over-emphasizing the negative results, the distribution of information between the main text and the supplement make the main text very hard to follow without reference to the supplement. Several of the supplemental figures would be better in the main text. For example, the definition of the expressed regions in Supplementary Figure 1 is critical for understanding results shown in the main text figures. The supplementary figures often make the same points as the main text, but illustrate them better. In addition to Supplementary Figures 1, Supplementary Figures 5-9 would all be better in the main text.

(5) In the Results section, a great deal of space is given over to descriptions of the topology, which can be discerned from the figures, rather than information about what model was used for molecular replacement: the fact that an alpha-Fold model was used merits mention here rather than just in the Methods section.

(6) The Discussion re-summarizes results, including the negative ones, but does not provide new insights. Only the final sentences of the third and fourth paragraphs present implications of the results for future work.

(7) There are many missed opportunities for insights, such as what was the previous evidence for an alternative binding site, comparison with another galactose-binding beta-trefoil lectin E. histolytica lectin [reference 26], the unusual and intriguing role of the glycan at a domain interface, and the accessibility of the sugar-binding site for interaction with large mucin molecules in the context of the whole protein, the membrane, and the flexible arm.

(8) There are some significant misstatements and the writing is repetitive, with roughly ten sentences beginning "Indeed". The title is somewhat misleading in referring to "the" Gal/GalNAc-binding lectin, given that there is at least one other such lectin as noted above. The statement on page 5 that the beta-trefoil is preceded by a region "without secondary structure" suggests a disordered region, while in fact a region without regular secondary structure elements is meant. The term "structural homology" is used in several places where structural similarity is meant. The significance of some details provided in the Methods section, such as the fact that NMR tubes were "placed in 96-tube racks", is unclear. On page 11, PDB entry 6IFB is cited as a model used for positioning galactose in the binding site, but this entry contains rhamnose, not galactose. The legend to Figure 3 refers to a Panel E that does not seem to exist.

Reviewer #3: All the points stated below could be included in the supplementary materials or discussion/results as appropriate.

1. Elaborate all the methods with inclusion of data from controls, protein purification/extraction, protein extraction/purification PAGE/Western blot images. Provide the details of Purification reagents, detergents etc/buffers for extraction of native hetero-dimer.

2. Show detailed binding sites for LacNAc on LGL. Discuss if data suggests that LacNAc and Gal-GalNAc will have similar affinity based on the structure. Do HGL atoms interact with the carbohydrate atoms?

3. Provide coulomb density map for the LacNAc binding site including the binding site residues which are within 6A (0.6nm) from any of the LacNAc atoms. Provide the same without modelling LacNAc into the density. At what sigma level the coulomb data was used to model the carbohydrate?

4. Provide NMR raw data (1D H NMR) highlighting the Amino (backbone) and Methyl (Core) shifts to provide support for your statement that the extracted/purified proteins are properly folded.

5. Discuss if the symmetry related molecules in the crystal structure of HGL_03 has impact on carbohydrate binding.

6. Authors have used Cys-Ser mutant for Hgl_03, discuss if this could impair loss in carbohydrate binding, given the fact that earlier CRD characterization studies (referred in this manuscript) used wild type Cys containing proteins. If oligomerization of HGL_03 is required to constitute the carbohydrate binding site, this may be feasible.

7. Do the cryo-electron microscopic data account for the trans-membrane and cytosolic part of the complex? elaborate.

8. The long standing experimental outcomes in the amoeba field depicts HGL to harbour the CRD, however the current study instead pointing towards LGL, could there be possibility of both hoarbouring the sites with very dissimilar in vitro affinities but still physiologically relevant in their capacity to carbohydrate binding in the host gut during amoebic infection.

PLOS authors have the option to publish the peer review history of their article (what does this mean? ). If published, this will include your full peer review and any attached files.

**Do you want your identity to be public for this peer review?** For information about this choice, including consent withdrawal, please see our Privacy Policy .

Reviewer #1: No

Reviewer #2: No

Reviewer #3: **Yes:** Sunando Datta

**Figure resubmission:**

**Reproducibility:**



---

## [Editor Report · Decision Letter 1]

28 Jan 2026

Dear Prof Higgins,

We are pleased to inform you that your manuscript 'Structural basis for carbohydrate recognition by the Gal/GalNAc lectin of Entamoeba histolytica involved in host cell adhesion' has been provisionally accepted for publication in PLOS Pathogens.

Best regards,

Dominique Soldati-Favre

Section Editor

PLOS Pathogens

Dominique Soldati-Favre

Section Editor

PLOS Pathogens

Sumita Bhaduri-McIntosh

Editor-in-Chief

PLOS Pathogens

orcid.org/0000-0003-2946-9497

Michael Malim

Editor-in-Chief

PLOS Pathogens

orcid.org/0000-0002-7699-2064
---

## [Editor Report · Acceptance letter]

Dear Prof Higgins,

We are delighted to inform you that your manuscript, "Structural basis for carbohydrate recognition by the Gal/GalNAc lectin of Entamoeba histolytica involved in host cell adhesion," has been formally accepted for publication in PLOS Pathogens.

Best regards,

Sumita Bhaduri-McIntosh

Editor-in-Chief

PLOS Pathogens

orcid.org/0000-0003-2946-9497

Michael Malim

Editor-in-Chief

PLOS Pathogens

orcid.org/0000-0002-7699-2064